# Investigating the Origin and Evolution of Polyploid *Trifolium medium* L. Karyotype by Comparative Cytogenomic Methods

**DOI:** 10.3390/plants12020235

**Published:** 2023-01-04

**Authors:** Eliška Lukjanová, Alžběta Hanulíková, Jana Řepková

**Affiliations:** Department of Experimental Biology, Faculty of Sciences, Masaryk University, 611 37 Brno, Czech Republic

**Keywords:** clover, zigzag clover, fluorescent in situ hybridization, 5S rDNA, 26S rDNA, centromeric repeat, polyploidy

## Abstract

*Trifolium medium* L. is a wild polyploid relative of the agriculturally important red clover that possesses traits promising for breeding purposes. To date, *T. medium* also remains the only clover species with which agriculturally important red clover has successfully been hybridized. Even though allopolyploid origin has previously been suggested, little has in fact been known about the *T. medium* karyotype and its origin. We researched *T. medium* and related karyotypes using comparative cytogenomic methods, such as fluorescent in situ hybridization (FISH) and RepeatExplorer cluster analysis. The results indicate an exceptional karyotype diversity regarding numbers and mutual positions of 5S and 26S rDNA loci and centromeric repeats in populations of *T. medium* ecotypes and varieties. The observed variability among *T. medium* ecotypes and varieties suggests current karyotype instability that can be attributed to ever-ongoing battle between satellite DNA together with genomic changes and rearrangements enhanced by post-hybridization events. Comparative cytogenomic analyses of a *T. medium* hexaploid variety and diploid relatives revealed stable karyotypes with a possible case of chromosomal rearrangement. Moreover, the results provided evidence of *T. medium* having autopolyploid origin.

## 1. Introduction

Zigzag clover, *Trifolium medium*, is a wild species of the clover genus, itself one of the largest genera in the bean family (Fabaceae). This herbaceous species is native to southwestern Asia and Europe but can be found in Northern America and rather infrequently in the southern hemisphere, such as in Australia, New Zealand, and Africa. The clover genus is of general agronomic importance due to its ability to form a symbiotic relationship with nitrogen-fixing bacteria that results in protein-rich forage even as it decreases the need for nitrogen fertilizers [1]. A number of clover species are cultivated extensively today as fodder plants and/or green manure crops to reduce nutrient runoff and soil erosion [2]. The economically most important clovers, namely red clover (*Trifolium pratense* L.) and white clover (*Trifolium repens* L.), are the subjects of breeding programs to improve agronomic traits such as yield, longevity, and nutrient content while overcoming constraints such as proneness to specific diseases, pests, or abiotic stress factors.

The low persistency of red clover (generally 2–3 years) is even more impacted by intensive grazing systems [3]. Being a highly complex trait, persistency is difficult to address even by modern improvement methods based upon molecular genetics [4]. Introduction of traits associated with perennialism, such as rhizomatous habit, from more persistent clover species has been pursued for many years, as reviewed by Abberton [5]. However, the only stable and viable progeny was produced from hybrids between tetraploid *T. pratense* cv. Tatra as a female and octoploid *T. medium* as a male parent, despite the fact that those two species have a different basic chromosome number in the haploid state (x = 7 in *T. pratense* and x = 8 in *T. medium*) [6]. Variability of the resulting progeny has been evaluated at the genetic level by means of flow cytometry measurement of DNA content in the fifth hybrid generation and by evaluation of chromosome numbers and rDNA loci [7,8]. Additionally, morphological, reproductive, and agronomic traits have been evaluated [9,10]. The results of analyses of generations after several rounds of backcrossing with *T. pratense* have revealed a tendency to stabilize the genomes into the maternal state with 2n = 28, which was previously reported in other crop plants [11,12], as well as a significant proportion of genetic variability maintained, with chromosome numbers ranging from 2n = 22 to 2n = 47 [8]. Moreover, *T. medium* displays higher tolerance to cold and virus-caused diseases and high content of phytoestrogens [9]. Therefore, *T. medium* is a promising target for breeding purposes and, as the success of interspecific hybridization to a large extent reflects the compatibility of both parental genomes, it is essential to have broad knowledge of its genome.

The genome sequence of *T. medium* was published in 2014, and, to date, that remains the largest *Trifolium* sequencing project, as it has an estimated genome size of 3154 Mbp [13,14]. Only a partial genomic sequence of 492.7 Mb was assembled, however, due to fragmentation caused by the large haploid genome size, polyploid state, cross-pollination, and high fraction of repetitive elements. Because of the fragmentation, the assembly was not sufficient for purposes of comprehensive annotation and so the authors provided a characterization of repeat content and its comparative analysis between *T. medium* and *T. pratense* [13]. Even though the authors managed to fully annotate 46.74% of analysed sequences as repetitive elements, it is likely that, due to underestimation caused by the low number of reads used in the analysis (only 0.1× coverage), the overall repeat content may make up as much as to 70% of the genome.

The comparative analysis between *T. medium* and *T. pratense* repeat content revealed a significant expansion of Ty-3/Gypsy retrotransposons in the *T. medium* genome (6.65% in *T. pratense* and 26.29% in *T. medium*) and oppositely a reduction in Ty-1/Copia retrotransposons and DNA transposons in general (12.22% in *T. pratense* and 7.80% in *T. medium* and 6.07% in *T. pratense* and 2.89% in *T. medium*, respectively). The striking difference in Ty-3/Gypsy retrotransposons content and, in particular, expansion of chromovirus lineage, which in absolute numbers covers 54 Mb in *T. pratense* and more than 766 Mb *T. medium*, is presumably the main cause of the genome expansion in *T. medium* [13].

In *T. pratense*, a partial karyotype and cytogenetic map was constructed based on 5S rDNA, 26S rDNA, and initially seven (later fourteen) bacterial artificial chromosome probes containing microsatellite markers with known position [15,16]. The proposed bacterial artificial chromosomes allow identification of individual *T. pratense* chromosomes. This was due to the well-described genome of *T. pratense* with a genome assembly at a pseudo-molecule level [17,18].

The *T. medium* genome characterization resulted in successful validation of 45 *T. medium*-specific repetitive elements spanning 2.83% of its genome in total with newly discovered four repeats, namely two centromeric, one pericentromeric, and one subtelomeric. Based upon these newly discovered repetitive elements and their localization, determined by fluorescent in-situ hybridization (FISH), as well as previously reported 5S and 26S rDNA loci, a partial *T. medium* (2n = 8x = 64) karyotype was proposed [8,13]. 5S and 26S rDNA loci were separated on individual chromosomes with 12 chromosomes bearing 5S and eight bearing 26S rDNA loci. Ten and six chromosomes with 5S and 26S loci, respectively, were carrying also centromeric repeats TrM378 and TrM300. Centromeric repeats were observed on half of the chromosomes. In 24 chromosomes TrM378 was prevalent, while TrM300 prevailed in eight chromosomes. Subtelomeric repeat TrM179 was observed on one arm of 24 chromosomes, on 8 chromosomes co-localized with centromeric repeats TrM378 and TrM300, on 4 chromosomes with pericentromeric repeat TrM60, and on 12 chromosomes separately. Twelve chromosomes bore no analysed cytogenetic marker. Based on the hybridization pattern, the chromosomes were sorted into 11 categories. Moreover, based upon the presence of identified centromeric repeats on only half of the chromosomes, the authors hypothesized for *T. medium* an allopolyploid origin [13].

Diversity was described on the *T. medium* ploidy level, with total chromosome counts ranging between 2n = 6x = 48 (*T. medium* var. *sarosiense*), 2n = 8x = 64, and 2n = 10x = 80 [13,14,19]. Regardless of ploidy level, however, these published *T. medium* ecotypes have a coincident basic chromosome number x = 8, which also has been suggested to be the ancestral state for the whole genus [20]. Moreover, basic chromosome numbers x = 7, 6, and 5 have to date been found also in 31 *Trifolium* spp. [20].

On the basis of chloroplast *trnL* intron sequences, the *Trifolium* genus has been divided into the two subgenera *Chronosemium* and *Trifolium*, with the latter being subdivided into eight sections, and a phylogenetic tree for all subgenera and sections has been constructed [20]. *T. medium* belongs to the most numerous *Trifolium* subgenera section *Trifolium* (73 species), and it is closely related to a number of diploid species, including *T. pratense*, the only species with which *T. medium* has been successfully hybridized (Figure 1) [9].

The aims of this study were to use selected cytogenetic markers and available sequences to (i) evaluate karyotype diversity in various *T. medium* ecotypes, varieties, and related species; (ii) capture events accompanying karyotype evolution in subspecies and ecotypes with different ploidy level; and then (iii) confirm or refute the hypothesis that *T. medium* is of allopolyploid origin.

## 2. Results

### 2.1. Cytogenetic Variability in T. medium Ecotypes and Varieties

To evaluate cytogenetic variability in *T. medium* populations, we investigated seven *T. medium* ecotypes collected from different localities in the Czech Republic and the Republic of Serbia and four varieties obtained from breeding and research facilities. Chromosome numbers as well as numbers and positions of rDNA loci and loci of centromeric repeats TrM378 and TrM300 were evaluated by FISH (Table 1). Mitoses with hybridization signals based upon which the Table 1 were constructed can be found in the Appendix A (Appendix A).

Numbers of chromosomes ranged in analysed ecotypes and varieties between 64, 70, 74 and 76 and were constant in all evaluated mitoses except for *T. medium* ecotype 1, where chromosome counts were 64 and 74. Numbers and mutual positions of rDNA loci varied between 8 and 19 for 5S rDNA and from 7to 13 for 26S rDNA. In contrast to the number of chromosomes, significant variability was observed between evaluated mitoses. Constant numbers and positions of rDNA loci were observed in *T. medium* ecotypes 2, 4, and 5 and variety Ruža. Co-localization of all 26S rDNA loci with 5S rDNA loci in all analysed mitoses was observed only in *T. medium* ecotype 5 (Figure 2B). On the contrary, strict separation of all rDNA loci on different chromosomes was observed in *T. medium* ecotype 2 and *T. medium* varieties 8/40, Melot, and Ruža (Figure 2A).

As in the case of rDNA loci, numbers and mutual positions of centromeric repeats TrM378 and TrM300 loci fluctuated among analysed species as well as between mitoses evaluated for each ecotype and variety prepared from bulked root tips of several seedlings of the same accessions. Numbers of centromeric repeats loci ranged between 16 and 38 for TrM378 and from six to 23 for TrM300 (Table 1). In each analysed ecotype and variety, at least two different hybridization patterns were observed regarding numbers and/or mutual positions of TrM378 and TrM300 loci. Strict co-localization of all TrM300 loci with TrM378 loci in all mitoses was observed only in *T. medium* ecotype 5 (Figure 3B). Strict separation of centromeric repeats loci TrM378 and TrM300 on different chromosomes in all mitoses was observed in *T. medium* ecotype 2 (Figure 3).

### 2.2. T. medium var. sarosiense

*T. medium* var. *sarosiense* is a hexaploid variety with 2n = 6x = 48. Like *T. medium*, *T. medium* var. *sarosiense* partial karyotype was proposed based upon the hybridization pattern of *T. medium*-specific repetitive elements, namely two centromeric repeats, one pericentromeric repeat, and one subtelomeric repeat and rDNA loci (Figure 4 and Figure 5).

The 5S and 26S rDNA loci were separated on individual chromosomes, with ten chromosomes bearing 5S and six chromosomes bearing 26S rDNA loci. TrM378 and TrM300 were observed on eight chromosomes co-localized with 5S rDNA loci. All chromosomes carrying 26S rDNA loci carried also centromeric repeats TrM378 and TrM300. Correspondingly to *T. medium*, centromeric repeats were observed on half of the chromosomes, but TrM378 prevailed on all chromosomes bearing 5S and 26S rDNA loci (14) and TrM300 predominated on chromosomes without rDNA loci (ten). Expansion of subtelomeric repeat TrM179 compared to *T. medium* was observed with 38 chromosomes bearing signals. In contrast with *T. medium*, subtelomeric repeats were observed on all chromosomes bearing 5S rDNA loci, with one chromosome having two loci on both arms. Pericentromeric repeat TrM60 was not observed on *T. medium* var. *sarosiense* chromosomes. Only one chromosome pair bore no analysed cytogenetic marker.

### 2.3. Diploid Relatives of T. medium–T. alpestre, T. rubens and T. pignantii

For further analyses, three diploid species were selected on the basis of the phylogenetic tree for section *Trifolium* constructed by Ellison et al. [20] (Figure 1), namely *T. alpestre*, *T. pignantii*, and *T. rubens*. All these species are diploid with 2n = 2x = 16. Their partial karyotypes were proposed using *T. medium*-specific repetitive elements and rDNA loci as FISH probes (Figure 6, Figure 7 and Figure 8).

While all analysed species carried two 26S rDNA sites per diploid genome and centromeric repeats TrM378 and TrM300 were observed on half of the chromosomes, with TrM378 prevalent on six and TrM300 on two, great variability was observed in the hybridization pattern regarding the number and localization of 5S rDNA loci and telomeric repeat TrM179.

In *T. alpestre*, variability in 5S rDNA loci number was observed, as previously reported by Vozárová et al. [21], with 10 or 11 chromosomes carrying 5S rDNA signals. In *T. pignantii*, 5S rDNA loci were observed on all chromosomes. In both *T. alpestre* and *T. pignantii*, co-localization of 26S and 5S rDNA loci on one chromosome pair was observed and subtelomeric repeat TrM179 was observed on 12 chromosomes, with 2 chromosomes (chromosomal pair with 26S rDNA loci) carrying two loci on both arms. The latter signals were very weak, however, and difficult to capture. Centromeric repeats TrM378 were in three chromosome pairs co-localized with telomeric repeat TrM179.

In *T. rubens*, on the other hand, two chromosomes carried 5S rDNA signals separately and no co-localization with 26S rDNA loci was observed. Subtelomeric repeat TrM179 was observed on all chromosomes with only one locus on one chromosome arm and all signals were clearly visible, similarly to those in *T. medium* and *T. medium* var. *sarosiense*. In all analysed species, all chromosome pairs carried at least one of the analysed cytogenetic markers.

### 2.4. RepeatExplorer rDNA Cluster Analysis

Furthermore, *T. medium* 5S rDNA genomic organization and homogeneity were investigated using RepeatExplorer2/TAREAN clustering pipeline and cluster graph computation methods (Figure 9).

RepeatExplorer2/TAREAN clustering analyses revealed a simple graph organization with intergenic spacer of one origin (Figure 9, grey nodes). Number of reads in the cluster was 2061 and the genome proportion was 0.19% with consensus repeat size length 338 bp. The analysed cluster had k-mer coverage 0.861 with connected component index C of 0.994. A cluster annotation report can be found in the Appendix A (Appendix A).

## 3. Discussion

The Chromosome Count Database (CCDB) compiles 82 entries for *T. medium* L., ten of which are more recent than 2000 [22]. The recent chromosome numbers listed in the CCDB include 64, 72, 76, 79, 80, 82, 84, ca 96–98, and ca 126. In our study, the number of chromosomes in analysed *T. medium* ecotypes and varieties varied between 64, 70, 74 and 76, with the most common being 64 (corresponding to octoploid state.) The number of chromosomes in octoploid ecotypes and varieties corresponds with basic chromosome number x = 8, which agrees with the ancestral basic chromosome number suggested by Ellison et al. [20]. The uniformity in all analysed mitoses of specific ecotypes and varieties, with the exception of *T. medium* ecotype 1, in which chromosome counts varied between 64 and 74, and *T. medium* variety Ruža, in which seeds from specifically octoploid plants were obtained, indicates an overall tendency towards stabilization of chromosome numbers in genotypes originating from both natural environments and breeding programmes. As the microscope slides were prepared from root tips harvested from different seedlings, different chromosome numbers between mitoses probably reflects variation between genotypes. Overall, the variability in chromosome counts in *T. medium* ecotype 1 and *T. medium* variety Ruža might be caused by recent hybridization events with parental species of different chromosome numbers.

However, great variability has been observed in numbers of 5S and 26S rDNA loci between analysed ecotypes and varieties, as well as in their mutual positions. rDNA sites are predominant targets of repeated recombination events and frequently are targeted by mobile element insertions, possibly promoting changes in the number of rDNA loci numbers [23,24]. The numbers and locations of 5S rDNA loci are regarded as highly conserved in different plants [25,26,27]. On the other hand, 45S rDNA repeats including 26S are fragile and associated with epigenetic modifications, and high variability in number of 26S rDNA sites has been described in many plant species [25,28,29,30]. In our research, similarly to other *Trifolium* spp., diversification of 5S rDNA loci per haploid genome was more common than was diversification of 26S rDNA [21]. Polyploid karyotypes of *T. medium* ecotypes and varieties and natural hybridization events might have promoted deregulation of transposable elements as well as translocation events resulting in amplification of 5S rDNA sites [24,31].

Vozárová et al. [21] suggested for ancestral *Trifolium* karyotype x = 8 a single pair of 5S and 26S per haploid genome on separate chromosomes. Regarding 5S rDNA loci, this possible ancestral constitution has been observed to be occurring in all analysed mitoses only in *T. medium* ecotype 2 and *T. medium* variety Ruža. These were the only ones of all analysed ecotypes and varieties that strictly follow the proposed ancestral state of one locus of both 5S and 26S rDNA per haploid genome. Variability in other ecotypes and varieties, including differentiation of loci numbers and their mutual positions, suggests substantial expansion, loss, and/or rearrangements and, similarly to *T. medium* clone 10/8 analysed by Dluhošová et al. [13], neither of these ecotypes nor varieties with octoploid chromosome count can be counted as true octoploids. Overall, variability in loci numbers between analysed mitoses of the same ecotype indicates ongoing evolution towards future stabilization and might emerge, among other reasons, from processes following natural hybridization between populations. Intraspecific variability of numbers of rDNA loci has been observed in many plant species, such as among the genera *Brachypodium* [32], *Oryza* [33], *Phaseolus* [34], *Paphiopedilum* [35], and many others. This natural variability might be further diversified by a mechanism similar to that observed after intergeneric or interspecific hybridization described in *Triticum* × *Aegilops* and *Festuca* × *Lolium* hybrids [36,37] or *Tragopogon mirus* Ownbey [38].

Constancy of mutual positions of 5S and 26S rDNA loci has been observed in three of four analysed varieties, with all rDNA loci separated on different chromosomes corresponding to the ancestral state positions. Observed stability can be explained by controlled hybridization in breeding facilities and conservation of current state. However, variability between numbers of rDNA loci indicates that loci numbers are under constant evolution. On the other hand, in the *T. medium* clone 10/8 described by Dluhošová et al. [13], which is the female parent of *T. medium* varieties 8/40 and 8/41, constant numbers of rDNA loci were described with twelve 5S rDNA and eight 26S rDNA separately. Observed variability in numbers of rDNA loci within its progeny might result from introducing genetic material of the male parent and post-hybridization events.

Satellite DNA, in higher plants preferentially localized to centromeric positions, is the most dynamic component of the genome during evolution [39]. Abundance of centromeric repeats, therefore, shows remarkable variation among species, sometimes resulting in emergence of a single species-specific repeat family. Usually, a single family of species-specific centromeric repeat occupies centromeres of all chromosomes [40]. This has been observed in *Oryza* [41], some *Brassicaceae* [42], and *Medicago* [43]. Moreover, single centromeric repeat has been reported on all chromosomes of *T. medium* relative *T. pratense* [13]. Exceptional variability of satellite repeats has been reported in other genera in the *Fabeae* tribe, however, reflecting even the variety of centromeric repeats [44,45,46]. Ávila Robledillo et al. [40] identified and characterized a diverse set of 64 families of centromeric repeats in 14 analysed species in the genera *Lens*, *Vicia*, *Pisum*, and *Lathyrus*. Dluhošová et al. [13] described in *T. medium* clone 8/10 a stable presence of centromeric repeats TrM378 and TrM300 co-localized on half of the chromosomes (32 of 64), upon the basis of which they suggested a theory of allopolyploid origin of *T. medium*. On 24 chromosomes TrM378 showed stronger signals, while TrM300 prevailed on only 8 chromosomes. Our research revealed great variability between both centromeric loci and their mutual position. Accordingly, in all analysed ecotypes and varieties numbers of TrM378 prevailed, thus suggesting expansion of this specific centromeric repeat at the expense of TrM300. In conflict with *T. medium* clone 10/8, separation of TrM378 and TrM300 was observed more often. This suggests a rivalry between centromeric repeats resulting in elimination of the losing repeat. Chromosomes with no hybridization signals of analysed centromeric repeats might have so few of the repetitive sequences that fluorescently marked probes produce uncapturable signals, possess satellite-free centromeres, as observed in potato (*Solanum tuberosum* L. [47]), or bear other yet unidentified centromeric repeat(s). In the case that *T. medium* chromosomes without hybridization signal are satellite-free, the karyotype might represent a transition stage between repeat-free and repeat-based centromeres where repeat-based centromeres might have retrotransposon-derived origin, as reported in sugarcane [48]. Variability regarding number and mutual positions of centromeric repeats loci between analysed ecotypes, varieties, and even mitoses might result from continuous battle between these repetitive elements. A similar phenomenon has been observed in the related legume *Phaseolus* [49]. The authors hypothesize that two centromeric repeats identified in *Phaseolus* spp. are currently not fixed in the genome but will ultimately reach a stable state with one repeat dominating on all chromosomes. This theory would support the centromere-drive hypothesis proposed by Henikoff et al. [50], according to which the centromeric satellite DNA acts as a selfish element resulting in an evolutionary arms race between selfish centromeric DNA, which propagates through female asymmetric meiosis at the expense of the homologous chromosome, and its associated kinetochore proteins, which must undergo adaptive evolution to maintain the interaction with the centromeric sequences. The current state of knowledge suggests, however, that the association and co-evolution of satellite repeats with interacting proteins in plants is far more complex. The centromere drive may act only periodically or just in specific cases, and this question demands further study [40]. The mechanism in *T. medium* might be enhanced and amplified by its polyploid state and ever-ongoing genomic changes and rearrangements with the tendency to return to diploid state. Nevertheless, the character of centromeric regions in these *T. medium* chromosomes remains a mystery and requires further investigation.

Comparative cytogenetic studies among species of certain taxa using markers such as ribosomal rDNA loci, tandem repeats, and/or bacterial artificial chromosomes are common in plant research and have been proven to provide valuable insights into karyotype evolution while allowing capture of chromosomal rearrangements [51,52,53,54,55]. In legumes, comparative cytogenetic and cytogenomic karyotype analyses have been published for several taxa, including *Phaseolus* [49], *Medicago* and allied genera [56], *Senna* [57,58], *Arachis* [59], *Canavalia* [60], *Lupinus* [61], and *Pisum* [62]. For this reason, we expanded our cytogenomic analyses to hexaploid *T. medium* var. *sarosiense* and diploid relatives *T. alpestre*, *T. pignantii*, and *T. rubens*. Quite the opposite of the significant variability among *T. medium* ecotypes and varieties previously described has been observed in *T. medium* var. *sarosiense*, which showed a stable karyotype with a constant number of all analysed elements. All 5S and 26S rDNA loci were localized on separate chromosomes, just as in *T. medium* clone 10/8 described by Dluhošová et al. [13]. Moreover, number of 26S rDNA loci in both of these karyotypes corresponds to the proposed ancestral state of one per haploid genome [21]. Identically to *T. medium* clone 10/8, we also observed the presence of both centromeric repeats TrM378 and TrM300 on half of the chromosomes (24 of 48). These repetitive elements and their constitution, therefore, can be acquired from a common ancestor and variability observed in *T. medium* ecotypes and varieties can be triggered by evolution in separate populations and post-hybridization events. The prevalence of TrM378 on *T. medium* clone 10/8 and *T. medium* var. *sarosiense* supports the hypothesis of ever-ongoing battle between selfish repetitive DNA.

In contrast to the *T. medium* clone 10/8, only one chromosome pair of *T. medium* var. *sarosiense* bore no signal. This is caused by substantial amplification of subtelomeric repeat TrM179, which we observed on 38 of 48 chromosomes (24 of 64 in *T. medium* clone 10/8). Together with centromeres, subtelomeric regions are the places accounting for the majority of constitutive heterochromatin. They are regions with a high abundance of satellite DNA that can act as selfish DNA and enhance their own transmission. In contrast to the *T. medium* clone 10/8, we observed one chromosome pair bearing subtelomeric repeat TrM179 signal on both arms. A similar phenomenon regarding differentiation of loci number and chromosomal hybridization pattern of satellite repeat has been described in *Medicago* and allied species by Rosato et al. [56]. The authors traced the presence of an AT-rich satellite repeat 185–189 bp in length which they determined to be preferentially localized in subtelomeric and interstitial regions. The analysed satellite DNA hybridized not only to different numbers of chromosomes but also in different hybridization patterns, upon which basis the authors divided the chromosomes into six categories. Overall, our results suggest stable localization of TrM179 in subtelomeric regions. One chromosome pair bearing two loci on both chromosomal arms probably emerged from rearrangement, such as a translocation or rare event of centric fission, followed by common centric fusion.

Regarding rDNA loci, karyotypes of *T. alpestre* and *T. rubens* have been already described by Vozárová et al. [21]. Nonetheless, we are the first to report number and position of rDNA loci in *T. pignantii*. This species represents the same phylogenetic branch, as does *T. alpestre*, with which it shares one locus of 26S rDNA per haploid genome co-localized with 5S rDNA loci. However, 5S rDNA sites were observed on all chromosomes. As in *T. medium* var. *sarosiense*, amplification of subtelomeric repeat TrM179 has been found with *T. alpestre* and *T. pignantii* having one chromosome pair bearing two loci of this subtelomeric repeat. This chromosomes pair seems identical in both of these species regarding hybridization patterns of the cytogenetic markers used, including rDNA loci. Therefore, it is probable that this chromosome emerged in a common ancestor of these two species. The presence of two subtelomeric repeats loci on one chromosome pair of *T. medium* var. *sarosiense*, however, with different hybridization signal regarding rDNA loci compared to *T. alpestre* and *T. rubens*, raises a question whether the origin of this chromosome pair first included chromosomal rearrangement that resulted in a chromosome pair bearing two loci of this subtelomeric repeat in the common ancestor of these three species. In this case, a different hybridization pattern regarding 5S and 26S rDNA loci would have occurred because of another rearrangement, causing their co-localization in the common ancestor of *T. alpestre* and *T. pignantii*. Alternatively, the chromosome pair bearing two loci of subtelomeric repeat might have emerged twice during evolution in two phylogenetic branches (first, leading to *T. medium* var. *sarosiense* and, second, leading to *T. alpestre* and *T. pignantii*) because of rearrangement between some recombination hot spots. In case of the latter, a unique genomic and/or epigenomic landscape in these species might be the cause of recurrent chromosomal rearrangements with reuse of DNA breakpoints, as described in the evolution of *Triticae* [63].

In all analysed diploid species and all mitoses, presence of centromeric repeats TrM378 and TrM300 was observed on half of the chromosomes with prevalence of TrM378, just as in *T. medium* clone 10/8 and *T. medium* var. *sarosiense* [13]. Furthermore, presence of analysed centromeric repeats in stable constitution in *T. medium* clone 10/8, *T*. *medium* var. *sarosiense*, and in the three diploid relatives suggests it has been inherited on half of the genome from the common ancestor of all these species. This is in contradiction to the hypothesis of allopolyploid origin of *T. medium* based on the presence of centromeric repeats on only half of the chromosomes as proposed by Dluhošová et al. [13]. Presence of the same centromeric repeat on different species of the same subclade indicating its ancestral status has recently been reported in other species, including carrot and a related legume bean [49,53]. In some of the analysed taxa, centromeric repeat probe hybridized uniformly to all chromosomes while others displayed different hybridization patterns with variable numbers of chromosomes devoid of centromeric repeat signals. The ancestral state of centromeric regions in the *T. medium* subclade might follow the hypothesis proposed by Iwata-Otsubo et al. [49] that centromeres evolve from repeat-free to repeat-based. Whether the repeat-based centromeres then emerged on all chromosomes of a diploid ancestor of all analysed taxa and over the course of time there was a distinction leading to loss or replacement of the repeat, as suggested in carrot [53], or whether the repeat-based centromeres appeared on only half of the chromosomes remains unclear.

Last but not least, Garcia et al. [64] suggested that the RepeatExplorer graph topologies of 5S rDNA clusters reflect divergence and number of homologous gene families in allopolyploid genomes. This results in a simple circular graph species with a single gene family and more complex graphs consisting of two or more interconnected loops representing intergenic spacers in allopolyploid genomes. Our results, therefore, proved the autopolyploid origin of *T. medium*.

## 4. Materials and Methods

### 4.1. Plant Material and Chromosome Preparation

Table 2 lists all 13 *T. medium* ecotypes and varieties as well as 3 diploid-related species used in this study and their seed origin. Seeds were germinated on Petri dishes in a refrigerator at 8–10 °C for 24 h and then transferred at 23 °C. Seedlings were placed into pots with soil or with perlite supplemented with Murashige and Skoog medium [65]. Root tips were harvested directly from germinated seeds or from growing plants.

The microscope slides were prepared following protocols by Lysák and Mandáková [66] and by Kirov et al. [67], with only slight modifications. On the first day, root tips were harvested and pretreated in cold water overnight. The second day, the cold water was replaced by freshly prepared 96% ethanol and 99% acetic acid in a 3:1 ratio and left overnight. On the third day, root tips were transferred to freshly prepared 100% ethanol and 99% acetic acid in a 3:1 ratio and stored at −20 °C. For the preparation of microscope slides, root tips (1–2 per microscope slide) from several seedlings were bulked, washed in a 0.1 M citrate buffer (0.08 M sodium citrate dihydrate, 0.01 citric acid) and digested in 30 µL of enzymatic mixture containing 0.3% cellulase, 0.3% pectolyase, and 0.3% cytohelicase (Merck, Prague, Czech Republic) in a citrate buffer for 120–150 min at 37 °C. The cell suspension was vortexed, 470 µL of water was added, and the whole mixture was then centrifuged at 11,000 rpm (Eppendorf 5415C centrifuge, Hamburg, Germany) for 2 min. Supernatant was carefully removed, 470 µL of 96% ethanol was added, and the mixture was again centrifuged at 11,000 rpm for 1 min. The pellet was resuspended in ethanol (10 µL per slide). Ten microlitres of cell suspension in ethanol was dropped onto a slide and a few seconds later 20 µL of the first fixation (3:1 ethanol and acetic acid) was added. After 20–30 s, the slide was held upside down over steam from a water bath heated at 55–60 °C for 15–20 s. Then 5 µL of the second fixation (2:1 ethanol and acetic acid) was added and the slide was held upside down over the steam for 5 s. The slide was then dried at room temperature and stored in a refrigerator at 8–10 °C prior to further use. Those slides with suitable chromosome spreads were treated with 100 µg mL^−1^ RNase A (Sigma, St. Louis, MO, USA) in saline–sodium citrate (SSC) buffer (2× SSC; 0.3 sodium chloride, 30 mM trisodium citrate, pH 7.0) for 1 h at 37 °C in a humid chamber box, then with 0.1 µg mL^−1^ pepsin in 10 mM HCl for 5 min at 37 °C in a Coplin jar. Finally, they were washed two times in 2× SSC and dewatered in a 70–90–96% ethanol series. At least six microscope slides were prepared from different root tips for each analysed sample. The number of analysed mitoses on individual slides ranged between 1 and 10+.

### 4.2. Probe DNA Isolation and Labelling

The DNA probes were prepared (with slight modifications) following Vozárová et al. [21] for 26S rDNA and 5S rDNA loci and Dluhošová et al. [13] for centromeric repeats TrM378 and TrM300. Selected sequences were amplified by polymerase chain reaction (PCR). Primer sequences used for PCR can be found in Table 3.

The PCR mixture contained 1× GoTaq Reaction Buffer (Promega, Madison, WI, USA), 0.2 mM deoxyribonucleotide triphosphate, 1 µL primers, 0.5 U Taq Polymerase (Promega, Prague, Czech Republic), and 20 ng of gDNA (*T*. *pratense* var. Tatra for 5S rDNA and *Arabidopsis thaliana* for 26S rDNA, *T. medium* for TrM378, TrM300, and TrM179). PCR products were separated by electrophoresis in 1% agarose gel and the products of corresponding length were excised from the gel. The DNA was extracted and purified using Nucleospin Gel and PCR Clean-Up Kit (Macherey-Nagel, Düren, Germany) and further purified by precipitating in sodium acetate. The DNA yield was quantified using a NanoDrop 2000c spectrophotometer (Thermo Scientific, Waltham, MA, USA). Probes were labelled by nick translation using biotin and digoxigenin Nick Translation Mix (Roche, Mannheim, Germany) and Ulysis Alexa Fluor 594 Nucleic Acid Labelling Kit (Invitrogen, Waltham, MA, USA).

### 4.3. Fluorescence In Situ Hybridization

The hybridization mixture in volume 25 µL contained 50% formamide, 10% dextran sulphate, 2× SSC, and 1 µL of each of those probes used (maximum three probes at a time; 26S rDNA, 5S rDNA, TrM378, TrM300, TrM179) in a final concentration of 100 ng per used volume. The mixture was denatured at 96 °C for 10 min, then rapidly cooled for 2 min at −20 °C in a cooling block. The mixture was applied onto a chosen slide, co-denaturated on a hot plate at 80 °C for 2 min, then incubated overnight at 37 °C in a humid chamber box. Post-hybridization washing was carried out at 42 °C with the following steps: 2× SSC twice for 5 min, 10% formamide in 0.1× SSC twice for 5 min, 2× SSC for 5 min, and 4× SSC + 0.05% Tween-20 for 5 min. Biotin and digoxigenin-labelled probes were immunodetected using streptavidin-Cy3 (GE Healthcare, Buckinghamshire, UK; 1:750 dilution) and anti-DIG-FITC (Roche; 1:250 dilution) antibodies, respectively. Chromosomes were counterstained with 40,6-diamidino-2-phenylindole (DAPI) in Vectashield (Vector Laboratories, Burlingame, CA, USA).

Images were captured using an Olympus BX 51 (Olympus, Tokyo, Japan) fluorescence microscope equipped with an Olympus DP72 CCD camera. Three greyscale images were taken of each mitosis event, and images were pseudocoloured using Adobe Photoshop CS6 software.

In some cases, the slides were washed for re-hybridization of another probe mixture. In these cases, the slides with probes hybridized were washed at room temperature in the following steps: 2× SSC twice for 10 min, 4× SSC + 0.1% Tween three times for 45 min, then the first fixation (3:1 ethanol and acetic acid) for 60 min. The slides were left to dry naturally, and the new hybridization mixture was applied.

### 4.4. RepeatExplorer rDNA Clustering Analysis

The sequence reads of *T. medium* in fastq format were acquired from GeneBank (accession SRR3229323). Total number of sequence reads was 709,783,962, with length ranging from 19 to 99 bp. The fastq reads were pre-processed by Quality Control tools implemented in the RepeatExplorer2 pipeline for quality filtering and trimming to a uniform length of 81 bp [68]. The pipeline is implemented within the international ELIXIR infrastructure (European research infrastructure for biological information).

Fastq files were converted into fasta and the reads were then analysed with RepeatExplorer2 using default parameters. The RepeatExplorer 2 computational pipeline uses graph-based clustering of next-generation sequencing reads to assemble groups of frequently overlapping reads into clusters of reads. The pipeline utilizes a BLAST threshold of 90% similarity. The clusters then represent repetitive elements. The total number of analysed reads was 1,102,062. The Cluster annotation files were searched for 5S rDNA using “5S rDNA” as search keywords. Cluster graph topology was visually investigated.

## 5. Conclusions

*T. medium* ecotypes and varieties show a remarkable level of variability regarding the numbers and mutual positions of rDNA loci and loci of *T. medium*-specific centromeric repeats. This can be attributed to ever-ongoing genomic changes and rearrangements along with the tendency to return to diploid state enhanced by genomic disbalance such as deregulation of transposable elements following natural hybridization. No intraspecific variability has been observed between hybridization patterns of the cytogenetic markers used in *T. medium* var. *sarosiense* and *T. medium* diploid relatives *T. alpestre*, *T. pignantii*, and *T. rubens.* Uniform presence of centromeric repeats on half of the chromosomes together with results of RepeatExplorer 5S rDNA clustering analysis constitute the evidence presented for autopolyploid origin of *T. medium*.

## Figures and Tables

**Figure 1 plants-12-00235-f001:**
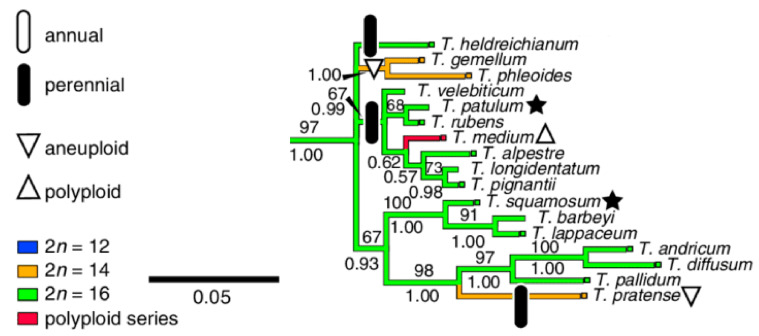
Branch of *Trifolium* section phylogenetic tree capturing *T. medium* and its closely related species (adapted from Ellison et al. [20]).

**Figure 2 plants-12-00235-f002:**
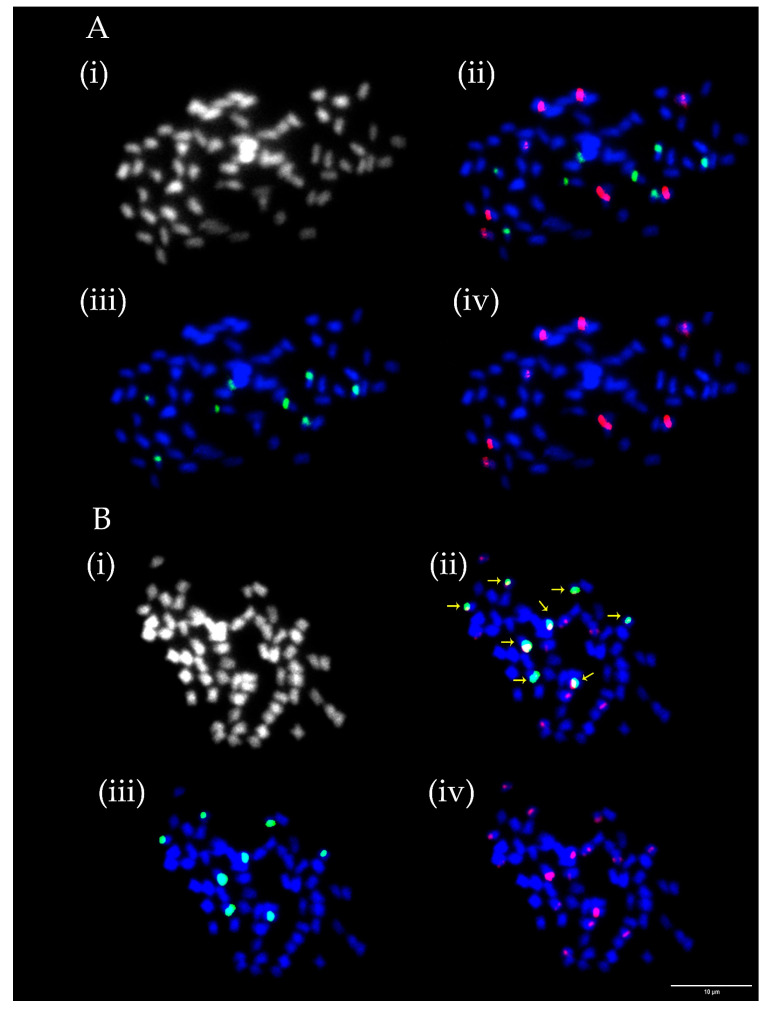
Numbers and localization of 5S (red) and 26S (green) loci on chromosomes of two *T. medium* ecotypes. (**A**) *T. medium* variety Ruža with eight 5S and eight 26S rDNA loci all occurring separately on different chromosomes. (**B**) *T. medium* ecotype 5 with 16 5S and eight 26S rDNA loci with all 26S rDNA loci co-localized with 5S rDNA loci. Four pictures are presented for each ecotype including chromosome spread (i) without fluorescent signals, (ii) with merged signals, (iii) with only 26S rDNA signals and (iv) with only 5S rDNA signals. Yellow arrows indicate *T. medium* ecotype 5 chromosomes with overlapping signals. Scale bar = 10 μm.

**Figure 3 plants-12-00235-f003:**
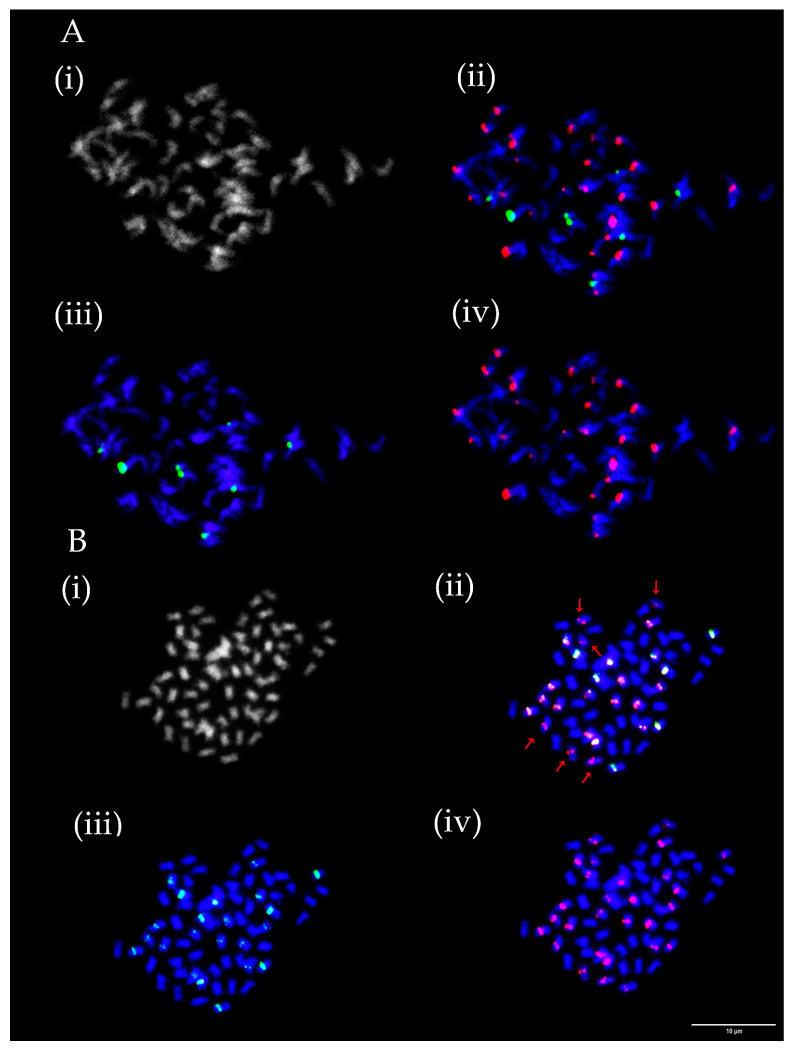
Numbers and localization of TrM378 (red) and TrM300 (green) loci on chromosomes of two *T. medium* ecotypes. (**A**) *T. medium* ecotype 2 with 16 TrM378 and 10 TrM300 loci, all separated on different chromosomes. (**B**) *T. medium* ecotype 5 with 28 TrM378 and 22 TrM300 loci with all TrM300 loci co-localized with TrM378 loci. Four pictures are presented for each ecotype including chromosome spread (i) without fluorescent signals, (ii) with merged signals, (iii) with only TrM300 signals and (iv) with only TrM378 signals. Red arrows indicate *T. medium* ecotype 5 chromosomes carrying only TrM378 signals. Scale bar = 10 μm.

**Figure 4 plants-12-00235-f004:**
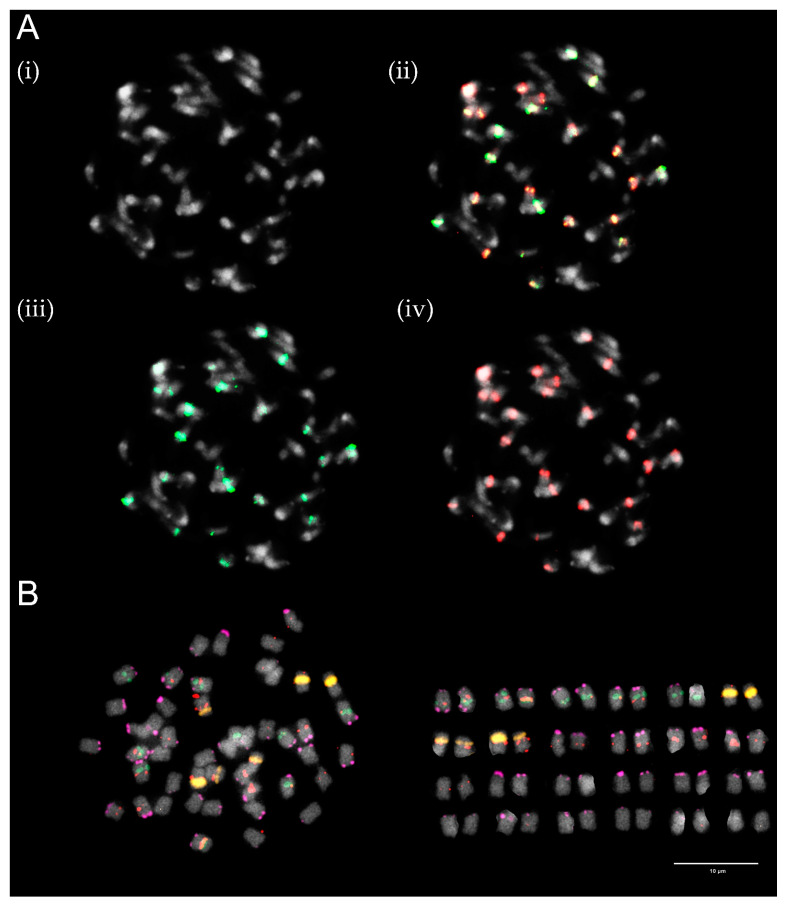
Hybridization pattern of analysed *T. medium*-specific repetitive elements on metaphase chromosomes of *T. medium* var. *sarosiense*. (**A**) Centromeric repeats TrM378 (red) and TrM300 (green). Four pictures are presented; (i) chromosome spread without fluorescent signals, (ii) with merged signals, (iii) with only TrM300 signals and (iv) with only TrM378 signals. (**B**) Centromeric repeat TrM378 (red), subtelomeric repeat TrM179 (pink), 5S rDNA (green), and 26S rDNA (orange). No presence of pericentromeric repeat TrM60 was observed. Scale bar = 10 μm.

**Figure 5 plants-12-00235-f005:**
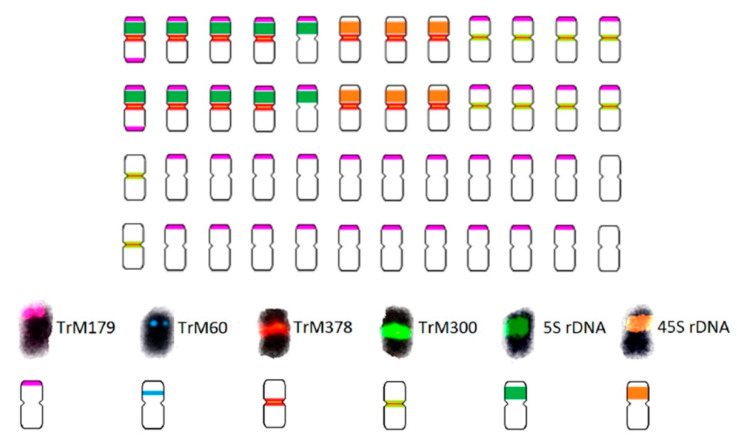
Schematic karyotype of *T. medium* var. *sarosiense* based on the hybridization pattern of probes derived from 5S (dark green) and 26S (orange) rDNA, subtelomeric repeat TrM179 (pink), pericentromeric repeat TrM60 (blue), and two centromeric repeats TrM378 (red) and TrM300 (light green). The ideogram illustrates numbers and mutual localizations of analysed cytogenetic markers only schematically and does not precisely capture specific chromosomal localization of each locus.

**Figure 6 plants-12-00235-f006:**
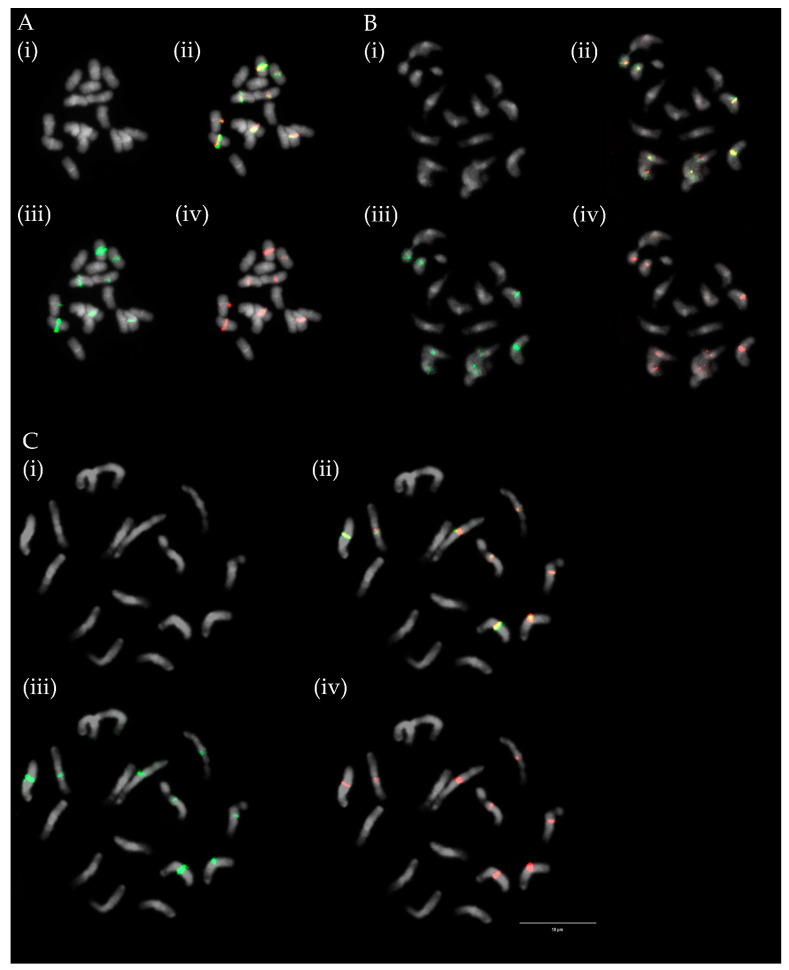
Hybridization patterns of centromeric repeats TrM378 (red) and TrM300 (light green) on metaphase chromosomes of (**A**) *T. alpestre*, (**B**) *T. pignantii*, and (**C**) *T. rubens*. Four pictures are presented for each species; (i) chromosome spread without fluorescent signals, (ii) with merged signals, (iii) with only TrM300 signals and (iv) with only TrM378 signals. Centromeric repeats were observed on half of the chromosomes of all analysed species, with TrM300 prevalent in one and TrM378 prevalent in three chromosome pairs. Scale bar = 10 μm.

**Figure 7 plants-12-00235-f007:**
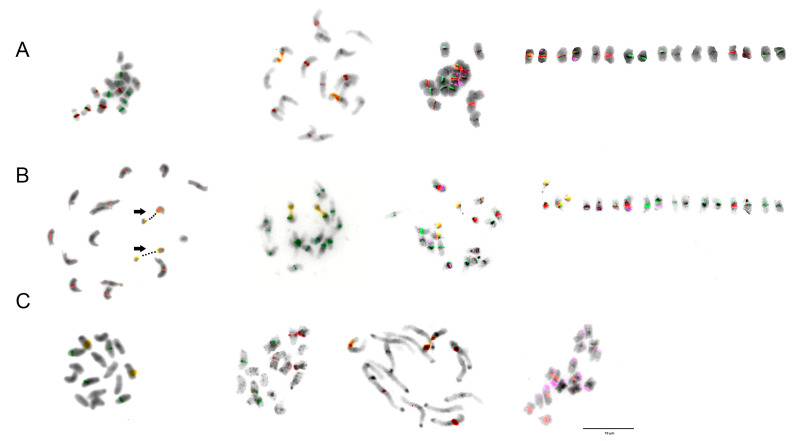
Hybridization patterns of selected *T. medium*-specific repetitive elements and 5S and 26S rDNA fluorescent probes on metaphase chromosomes of (**A**) *T. alpestre*, (**B**) *T. pignantii*, and (**C**) *T. rubens*. Centromeric repeat TrM378 (red), subtelomeric repeat TrM179 (pink), 5S rDNA (green), and 26S rDNA (orange). Black dots and black arrows indicate highlighted nucleolar organizer regions (NORs). Scale bar = 10 μm.

**Figure 8 plants-12-00235-f008:**
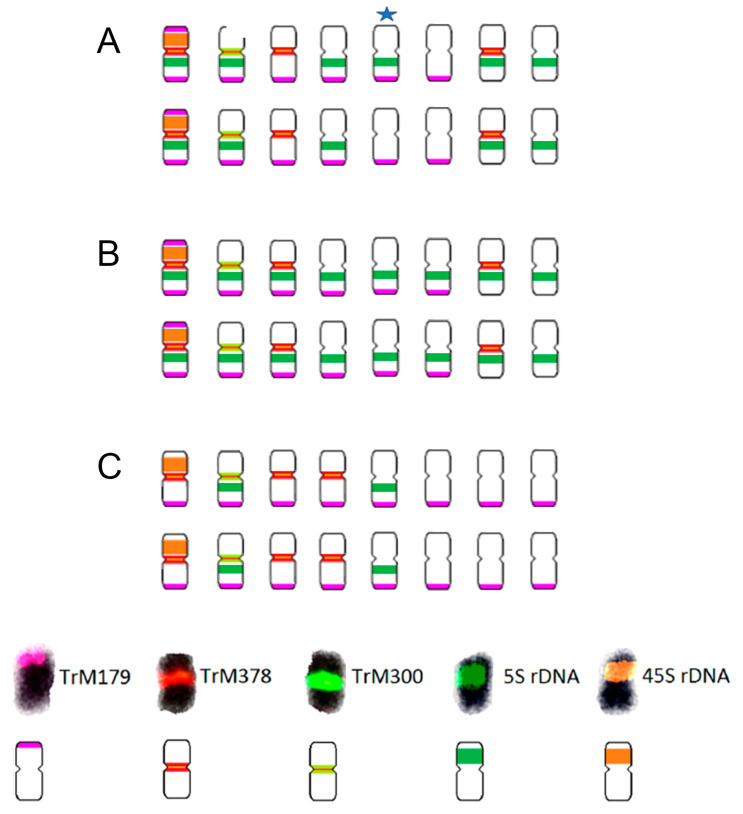
Schematic karyotypes of (**A**) *T. alpestre*, (**B**) *T. pignantii*, and (**C**) *T. rubens* based on the hybridization patterns of probes derived from 5S (dark green) and 26S (orange) rDNA; subtelomeric repeat TrM179 (pink); and two centromeric repeats, TrM378 (red) and TrM300 (light green). A star indicates an odd chromosome carrying 5S signal in only some mitoses prepared from individuals of the same accession. This odd chromosome carried subtelomeric repeat TrM179 in all analysed mitoses. The ideograms illustrate numbers and mutual localizations of analysed cytogenetic markers only schematically and do not precisely capture specific chromosomal localization of each locus.

**Figure 9 plants-12-00235-f009:**
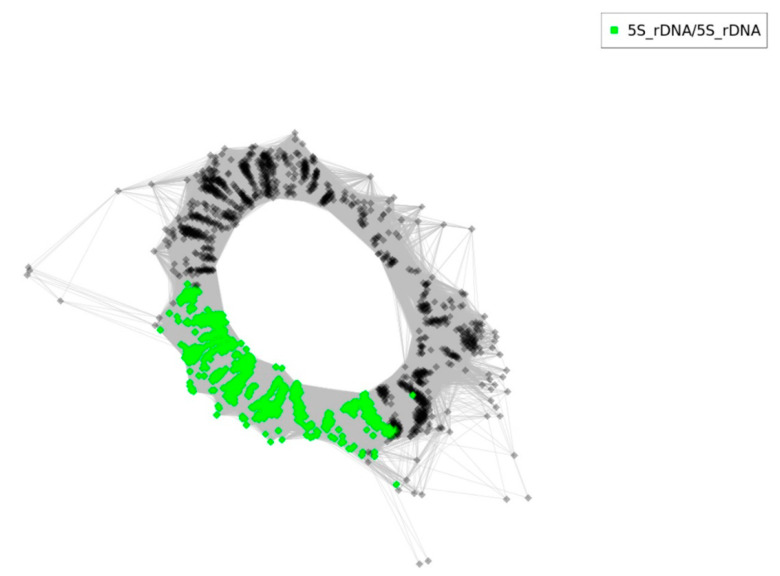
RepeatExplorer2 graphical output of *T. medium* 5S rDNA sequence reads cluster analysis. Nodes represent single reads. The intergenic spacers and 5S coding sequences are highlighted in grey and green, respectively.

**Table 1 plants-12-00235-t001:** Numbers of chromosomes, rDNA loci and centromeric repeats TrM378 and TrM300 loci and their positions in analysed *T. medium* ecotypes and varieties.

		Number of Chromosomes		5S rDNA	26S rDNA	26S rDNA Position			TrM378 Loci	TrM300 loci	TrM300 Position	
						Co-Localized with 5S	Separately				Co-Localized with TrM378	Separately
Ecotypes	*T. medium* 1	74		18	9	4	5		28	12	2	10
				19	9	7	2		28	10	4	6
		64		16	10	3	7					
	*T. medium* 2	64		8	8	0	all		24	8	0	all
									26	8	0	all
									16	10	0	all
	*T. medium* 3	64		15	8	all	0		22	20	11	9
				12	7	5	2		18	10	0	all
									16	9	0	all
	*T. medium* 4	64		13	7	1	6		18	8	0	all
									16	6	0	all
									16	10	2	8
	*T. medium* 5	64		16	8	all	0		24	8	all	0
									28	22	all	0
	*T. medium* 6	64		14	8	5	3		28	23	16	7
				10	8	4	4		20	7	4	3
				12	10	3	7					
	*T. medium* 7	70		14	10	5	5		22	10	8	2
				14	10	7	3		22	9	2	7
				18	8	7	1		24	12	0	all
Varieties	*T. medium* 8/40	64		10	7	0	all		38	18	6	12
				14	11	0	all		20	9	3	6
				10	10	0	all		32	9	0	all
	*T. medium* 8/41	64		8	6	0	all		28	8	6	2
				10	13	5	8		28	12	6	6
				10	8	0	all					
	*T. medium* Melot	76		10	12	0	all		32	10	0	all
				12	12	0	all		30	12	4	8
									32	8	2	6
	*T. medium* Ruža	64		8	8	0	all		26	12	6	6
									24	12	4	8

TrM378, TrM300—centromeric repeats.

**Table 2 plants-12-00235-t002:** Origin and accession numbers of analysed *Trifolium* species.

	Acquired from	Origin	Accession Number
*T. medium* 1	GRIN, CZ	CZ	13T0500049
*T. medium* 2	GRIN, CZ	CZ	13T0500437
*T. medium* 3	GRIN, CZ	SRB	13T0500062
*T. medium* 4	GRIN, CZ	SRB	13T0500110
*T. medium* 5	GRIN, CZ	SRB	13T0500108
*T. medium* 6	GRIN, CZ	CZ	13T0500116
*T. medium* 7	GRIN, CZ	CZ	13T0500113
*T. medium* 8/40 ^1^	RCGB, CZ	CZ	-
*T. medium* 8/41 ^1^	RCBG, CZ	CZ	-
*T. medium* Melot *^2^*	RIFC, Ltd., CZ	CZ	-
*T. medium* Ruža ^3^	RCGB, CZ	CZ	-
*T. medium* var. *sarosiense*	IPK, DE	-	TRIF 179
*T. alpestre*	IPK, DE	-	TRIF 210
*T. pignantii*	IPK, DE	-	TRIF 277
*T. rubens*	IPK, DE	GRE	TRIF 211

^1^ Sexual progeny of female *T. medium* clone 10/8 and unknown male. ^2^ Bred using wild ecotypes collected locally in the South Moravian Region of the Czech Republic. ^3^ Bred using *T. medium* varieties Melot and 10/8 and ecotypes from The U.S. National Plant Germplasm System Seed Bank, Beltsville, MD, USA and from the GeneBank of Crop Research Institute (GRIN), Prague-Ruzyně, Czech Republic. Seeds from plants with octoploid number of chromosomes were selected. IPK = Leibniz Institute of Plant Genetics and Crop Plant Research, Gatersleben, Germany. RCGB = Red Clover and Grass Breeding, Hladké Životice, Czech Republic. RIFC, Ltd. = Research Institute for Fodder Crops, Ltd., Troubsko, Czech Republic.

**Table 3 plants-12-00235-t003:** Primer sequences used for amplification of DNA for FISH probes.

Probe	Forward Primer	Reverse Primer	PCR Product Length (bp)
26S rDNA	TTCCCACTGTCCCTGTCTACTAT	GAACGGACTTAGCCAACGACA	899
5S rDNA	GGTGCGATCATACCAGCACTAA	GAGGTGCAACACAAGGACTTC	117
TrM378	ACTTTTGATCTGGTTATCTCT	ACTGTATATGAATCGAGAAGCA	378
TrM300	CTGTTAGTAAGCTATTAGAAGT	ATTTAACTTATCTGCACTATCTT	300
TrM179	CTCTACGTATTTCGGTAGTGCCC	TCATTGTTTTTACCCGACGAACG	132

## Data Availability

All data generated or analysed during this study are included in this published article. Further inquiries can be addressed to the corresponding author.

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
