# Peer review of "Investigating the Origin and Evolution of Polyploid Trifolium medium L. Karyotype by Comparative Cytogenomic Methods"

_plants, 2023, doi:10.3390/plants12020235_

Round 1
Reviewer 1 Report (Previous Reviewer 1)
All recomendations have been considered and necessary corrections were done
Author Response
Thank You for Your positive feedback. We are glad to see that in Your opinion the manuscript was improved.
Reviewer 2 Report (Previous Reviewer 2)
Authors have improved the figures 2 and 3, making unlikely that the 8 5S rDNA signals are results of misinterpretation of the highly intensive 26S rDNA signals. For the green signal most bottom left in figure 2B (iii) there is no (or only week) counterpart in 2B (iv). Also in 3B (iii) and 3B (iv) some of the signals are not of the expected intensity. So I can agree, that it is realy a biological pnenomenon.
Author Response
Thank You for Your feedback, we are happy to hear that our manuscript was improved according to Your opinion.
This manuscript is a resubmission of an earlier submission. The following is a list of the peer review reports and author responses from that submission.
Round 1
Reviewer 1 Report
The paper “Investigating the origin and evolution of polyploid Trifolium medium L. karyotype by comparative cytogenomic methods” by E. Lukjanova et al. presents new interesting data on karyotype organization, variability and evolution of wild polyploid relative of cultivated clover. These legume species has very small chromosomes with relatively low amounts of repeated sequences, which makes it difficult for cytogenomic analyses. The authors used several probes to characterize chromosomes and do karyotype comparisons. The topic of this paper fits the scope of MDPI Plants and can be considered for publications in this journal. There are however several claims and questions to authors which have to be responded.
First, text is very difficult to read, the manuscript needs language correction. For example: “In no species were constant numbers observed in all analyzed mitoses” (lines 155-156) - ? “carrying 26S rDNA loci centromeric repeats” (lines 186-187) – something is missing; How “diploid species closely related to T. medium were identified on the basis of the phylogenetic tree”? (lines 197-198); and many others.
Second, it is not clear from the description of methods whether chromosomal preparations were done from one seedling each or the roots from several seedlings were bulked and digested in enzyme solution? This is important point because the authors found variation in chromosome number from 64 to 74 in ecotype 1. Such variation can be due to variation between genotypes or variation within genotype.
Third, how many metaphase cells (per plant) have been scored to define chromosome number? In my mind, some cells shown on Figures and Supplementary materials are not scorable due to chromosome adhesions and numerous overlaps.
Reviewer 2 Report
The authors evaluated the karyotype diversity in various Trifolium medium (zigzag clover) ecotypes and compared it to that of varieties and related species to get insight into the evolution of this hexaploid species. The theme has some importance because T. medium id the only species successfully used as partner for interspecific crosses with the economically important red clover (T. pratense). The form of the manuscript is acceptable but it suffers by some weaknesses.
The key statement of the manuscript is an exceptional karyotype diversity regarding number and mutational positions of 5S and 26S rDNA loci. This should be demonstrated especially in Figure 2. Double hybridization with 5S (red-labeled) and 26S (green-labeled) probes in t. medium ecotype 2 and T. medium ecotype 5 mitotic chromosomes have been shown. For T. medium ecotype 5 (A) authors described 16 5S and 8 26S rDNA loci. By my opinion the large , slightly reddish signals (adjacent to the green signals) are due to an excessive intensity of the green signal and thereby a wave length penetration to the red signal spectrum. Consequently I would see in picture 2A) 8 signals of 26S rDNA (green) and 8 signals of 5S rDNA (red) and for this reason exactly the same result as for T medium ecotype 2 (2B). Such effects are well known in laboratory practice. If this is the case for the pictures shown in manuscript (mostly the best pictures available) I would have doubt for all other results. If the authors can show other pictures for ecotype 5 with both, clear red and green signals, on eight chromososomes I will correct myselfe and we can discuss further. Generally the quality of the cytological pictures is unsatisfactory. So in Figure 5B not more than 7 chromosomes show red and no more than 7 chromosomes show green signals.
In figure 3A) authors want to show, that all TrM300 signals are colocalized with TrM378. Also here I assume the same effect. Please repeat the hybridization with 26S rDNA under more stringent conditions or use an other combination of fluorescences/filters.
The cytogenetic analyses of T medium var. sarosiense and of the diploid relatives T. alpestre, T. rubens and T. pignanti inclusive the schematic karyotypes (Figures 5 and 8) seem to be more exact, but without the results of the first part of the results there is not enougth substance for publication in a journal as "plants".
The parts about GISH analyses (M&M, results and discussion) can be deleted because it was not successful.
Some small remarks: Table 1 and table 2 should be integrated into one table.
Sometimes you switch between 26S rDNA and 25S rDNA to describe your probe.
